# Fast and Flexible Multi-Task Classification Using Conditional Neural Adaptive Processes

**James Requeima**[*]
University of Cambridge
Invenia Labs
jrr41@cam.ac.uk

**Jonathan Gordon**[*]
University of Cambridge
jg801@cam.ac.uk

**John Bronskill**[*]
University of Cambridge
jfb54@cam.ac.uk

**Sebastian Nowozin**
Google Research Berlin
nowozin@google.com

**Richard E. Turner**
University of Cambridge
Microsoft Research
ret26@cam.ac.uk

## Abstract

The goal of this paper is to design image classification systems that, after an initial multi-task training phase, can automatically adapt to new tasks encountered at test time. We introduce a conditional neural process based approach to the multi-task classification setting for this purpose, and establish connections to the meta-learning and few-shot learning literature. The resulting approach, called CNAPs, comprises a classifier whose parameters are modulated by an adaptation network that takes the current task's dataset as input. We demonstrate that CNAPs achieves state-of-the-art results on the challenging META-DATASET benchmark indicating high-quality transfer-learning. We show that the approach is robust, avoiding both over-fitting in low-shot regimes and under-fitting in high-shot regimes. Timing experiments reveal that CNAPs is computationally efficient at test-time as it does not involve gradient based adaptation. Finally, we show that trained models are immediately deployable to continual learning and active learning where they can outperform existing approaches that do not leverage transfer learning.

## 1 Introduction

We consider the development of general purpose image classification systems that can handle tasks from a broad range of data distributions, in both the low and high data regimes, without the need for costly retraining when new tasks are encountered. We argue that such systems require mechanisms that adapt to each task, and that these mechanisms should themselves be learned from a diversity of datasets and tasks at training time. This general approach relates to methods for meta-learning [1, 2] and few-shot learning [3]. However, existing work in this area typically considers homogeneous task distributions at train and test-time that therefore require only minimal adaptation. To handle the more challenging case of different task distributions we design a fully adaptive system, requiring specific design choices in the model and training procedure.

Current approaches to meta-learning and few-shot learning for classification are characterized by two fundamental trade-offs. (i) The number of parameters that are adapted to each task. One approach adapts only the top, or head, of the classifier leaving the feature extractor fixed [4, 5]. While useful in simple settings, this approach is prone to under-fitting when the task distribution is heterogeneous [6]. Alternatively, we can adapt all parameters in the feature extractor [7, 8] thereby increasing

---

[*]Authors contributed equally

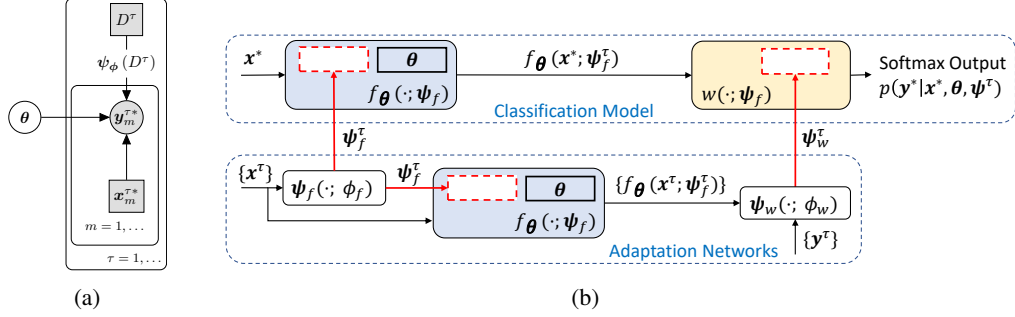

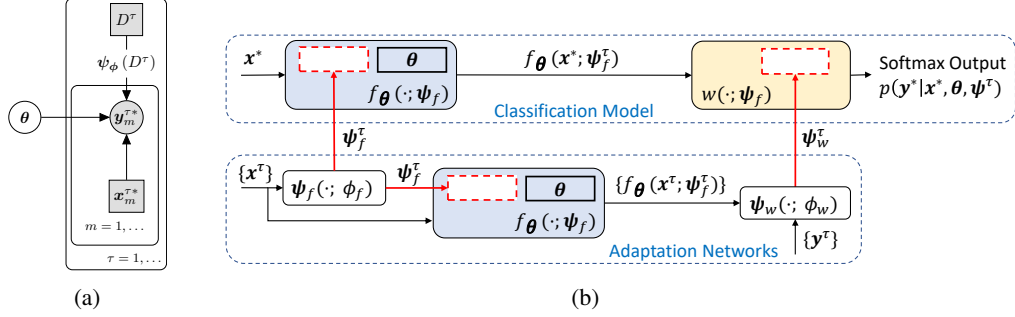

Figure 1: (a) Probabilistic graphical model detailing the CNP [13] framework. (b) Computational diagram depicting the CNAPS model class. Red boxes imply parameters in the model architecture supplied by adaptation networks. Blue shaded boxes depict the feature extractor and the gold box depicts the linear classifier.

fitting capacity, but incurring a computation cost and opening the door to over-fitting in the low-shot regime. What is needed is a middle ground which strikes a balance between model capacity and reliability of the adaptation. (ii) The adaptation mechanism. Many approaches use gradient-based adaptation [7, 9]. While this approach can incorporate training data in a very flexible way, it is computationally inefficient at test-time, may require expertise to tune the optimization procedure, and is again prone to over-fitting. Conversely, function approximators can be used to directly map training data to the desired parameters (we refer to this as *amortization*) [5, 10]. This yields fixed-cost adaptation mechanisms, and enables greater sharing across training tasks. However, it may under-fit if the function approximation is not sufficiently flexible. On the other hand, high-capacity function approximators require a large number of training tasks to be learned.

We introduce a modelling class that is well-positioned with respect to these two trade-offs for the multi-task classification setting called Conditional Neural Adaptive Processes (CNAPS).[2] CNAPS directly model the desired predictive distribution [11, 12], thereby introducing a *conditional neural processes* (CNPs) [13] approach to the multi-task classification setting. CNAPS handles varying way classification tasks and introduces a parametrization and training procedure enabling the model to *learn to adapt* the feature representation for classification of diverse tasks at test time. CNAPS utilize i) a classification model with shared global parameters and a small number of task-specific parameters. We demonstrate that by identifying a small set of key parameters, the model can balance the trade-off between flexibility and robustness. ii) A rich adaptation neural network with a novel auto-regressive parameterization that avoids under-fitting while proving easy to train in practice with existing datasets [6]. In Section 5 we evaluate CNAPS. Recently, Triantafillou et al. [6] proposed META-DATASET, a few-shot classification benchmark that addresses the issue of homogeneous train and test-time tasks and more closely resembles real-world few-shot multi-task learning. Many of the approaches that achieved excellent performance on simple benchmarks struggle with this collection of diverse tasks. In contrast, we show that CNAPS achieve state-of-the-art performance on the META-DATASET benchmark, often by comfortable margins and at a fraction of the time required by competing methods. Finally, we showcase the versatility of the model class by demonstrating that CNAPS can be applied "out of the box" to continual learning and active learning.

## 2 Model Design

We consider a setup where a large number of training tasks are available, each composed of a set of inputs $x$ and labels $y$. The data for task $\tau$ includes a *context set* $D^\tau = \{(x_n^\tau, y_n^\tau)\}_{n=1}^{N_\tau}$, with inputs and outputs observed, and a *target set* $\{(x_m^{\tau*}, y_m^{\tau*})\}_{m=1}^{M_\tau}$ for which we wish to make predictions ($y^{\tau*}$ are only observed during training). CNPs [13] construct predictive distributions given $x^*$ as:

$$p\left(y^*|x^*, \theta, D^\tau\right) = p\left(y^*|x^*, \theta, \psi^\tau = \psi_\phi\left(D^\tau\right)\right). \tag{1}$$

Here $\theta$ are global classifier parameters shared across tasks. $\psi^\tau$ are local task-specific parameters, produced by a function $\psi_\phi(\cdot)$ that acts on $D^\tau$. $\psi_\phi(\cdot)$ has another set of global parameters $\phi$ called *adaptation network parameters*. $\theta$ and $\phi$ are the learnable parameters in the model (see Figure 1a).

[2]Source code available at `https://github.com/cambridge-mlg/cnaps`.

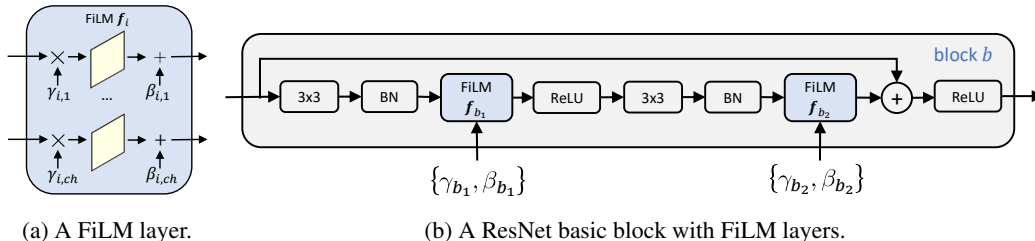

(a) A FiLM layer.        (b) A ResNet basic block with FiLM layers.

Figure 2: (Left) A FiLM layer operating on convolutional feature maps indexed by channel $ch$. (Right) How a FiLM layer is used within a basic Residual network block [14].

CNAPs is a model class that specializes the CNP framework for the multi-task classification setting. The model-class is characterized by a number of design choices, made specifically for the multi-task image classification setting. CNAPs employ global parameters $\boldsymbol{\theta}$ that are trained offline to capture high-level features, facilitating transfer and multi-task learning. Whereas CNPs define $\boldsymbol{\psi}^\tau$ to be a fixed dimensional vector used as an input to the model, CNAPs instead let $\boldsymbol{\psi}^\tau$ be specific parameters of the model itself. This increases the flexibility of the classifier, enabling it to model a broader range of input / output distributions. We discuss our choices (and associated trade-offs) for these parameters below. Finally, CNAPs employ a novel auto-regressive parameterization of $\boldsymbol{\psi_\phi}(\cdot)$ that significantly improves performance. An overview of CNAPs and its key components is illustrated in Figure 1b.

## 2.1 Specification of the classifier: global $\theta$ and task-specific parameters $\psi^\tau$

We begin by specifying the classifier's global parameters $\boldsymbol{\theta}$ followed by how these are adapted by the local parameters $\boldsymbol{\psi}^\tau$.

**Global Classifier Parameters**. The global classifier parameters will parameterize a feature extractor $f_{\boldsymbol{\theta}}(\boldsymbol{x})$ whose output is fed into a linear classifier, described below. A natural choice for $f_{\boldsymbol{\theta}}(\cdot)$ in the image setting is a convolutional neural network, e.g., a ResNet [14]. In what follows, we assume that the global parameters $\boldsymbol{\theta}$ are fixed and known. In Section 3 we discuss the training of $\boldsymbol{\theta}$.

**Task-Specific Classifier Parameters: Linear Classification Weights**. The final classification layer must be task-specific as each task involves distinguishing a potentially unique set of classes. We use a task specific affine transformation of the feature extractor output, followed by a softmax. The task-specific weights are denoted $\boldsymbol{\psi}_w^\tau \in \mathbb{R}^{d_f \times C^\tau}$ (suppressing the biases to simplify notation), where $d_f$ is the dimension of the feature extractor output $f_{\boldsymbol{\theta}}(\boldsymbol{x})$ and $C^\tau$ is the number of classes in task $\tau$.

**Task-Specific Classifier Parameters: Feature Extractor Parameters**. A sufficiently flexible model must have capacity to adapt its feature representation $f_{\boldsymbol{\theta}}(\cdot)$ as well as the classification layer (e.g. compare the optimal features required for ImageNet versus Omiglot). We therefore introduce a set of local feature extractor parameters $\boldsymbol{\psi}_f^\tau$, and denote $f_{\boldsymbol{\theta}}(\cdot)$ the *unadapted* feature extractor, and $f_{\boldsymbol{\theta}}(\cdot; \boldsymbol{\psi}_f^\tau)$ the feature extractor adapted to task $\tau$.

It is critical in few-shot multi-task learning to adapt the feature extractor in a parameter-efficient manner. Unconstrained adaptation of all the feature extractor parameters (e.g. by fine-tuning [9]) gives flexibility, but it is also slow and prone to over-fitting [6]. Instead, we employ linear modulation of the convolutional feature maps as proposed by Perez et al. [15], which adapts the feature extractor through a relatively small number of task specific parameters.

A Feature-wise Linear Modulation (FiLM) layer [15] scales and shifts the $i^{th}$ unadapted feature map $\boldsymbol{f}_i$ in the feature extractor FiLM$(\boldsymbol{f}_i; \gamma_i^\tau, \beta_i^\tau) = \gamma_i^\tau \boldsymbol{f}_i + \beta_i^\tau$ using two task specific parameters, $\gamma_i^\tau$ and $\beta_i^\tau$. Figure 2a illustrates a FiLM layer operating on a convolutional layer, and Figure 2b illustrates how a FiLM layer can be added to a standard Residual network block [14]. A key advantage of FiLM layers is that they enable expressive feature adaptation while adding only a small number of parameters [15]. For example, in our implementation we use a ResNet18 with FiLM layers after every convolutional layer. The set of task specific FiLM parameters ($\boldsymbol{\psi}_f^\tau = \{\boldsymbol{\gamma}_i^\tau, \boldsymbol{\beta}_i^\tau\}$) constitute fewer than 0.7% of the parameters in the model. Despite this, as we show in Section 5, they allow the model to adapt to a broad class of datasets.

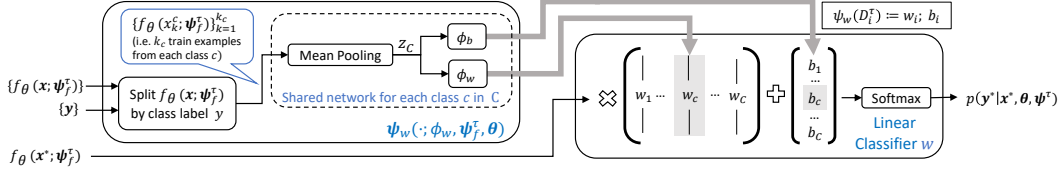

Figure 3: Implementation of functional representation of the class-specific parameters $\boldsymbol{\psi}_w$. In this parameterization, $\boldsymbol{\psi}_w^c$ are the linear classification parameters for class $c$, and $\boldsymbol{\phi}_w$ are the learnable parameters.

## 2.2 Computing the local parameters via adaptation networks

The previous sections have specified the form of the classifier $p\left(\boldsymbol{y}^* | \boldsymbol{x}^*, \boldsymbol{\theta}, \boldsymbol{\psi}^\tau\right)$ in terms of the global and task specific parameters, $\boldsymbol{\theta}$ and $\boldsymbol{\psi}^\tau = \{\boldsymbol{\psi}_f^\tau, \boldsymbol{\psi}_w^\tau\}$. The local parameters could now be learned separately for every task $\tau$ via optimization. While in practice this is feasible for small numbers of tasks (see e.g., [16, 17]), this approach is computationally demanding, requires expert oversight (e.g. for tuning early stopping), and can over-fit in the low-data regime.

Instead, CNAPs uses a function, such as a neural network, that takes the context set $D^\tau$ as an input and returns the task-specific parameters, $\boldsymbol{\psi}^\tau = \boldsymbol{\psi}_\phi\left(D^\tau\right)$. The adaptation network has parameters $\boldsymbol{\phi}$ that will be trained on multiple tasks to learn how to produce local parameters that result in good generalisation, a form of meta-learning. Sacrificing some of the flexibility of the optimisation approach, this method is comparatively cheap computationally (only involving a forward pass through the adaptation network), automatic (with no need for expert oversight), and employs explicit parameter sharing (via $\boldsymbol{\phi}$) across the training tasks.

**Adaptation Network: Linear Classifier Weights**. CNAPs represents the linear classifier weights $\boldsymbol{\psi}_w^\tau$ as a parameterized function of the form $\boldsymbol{\psi}_w^\tau = \boldsymbol{\psi}_w(D^\tau; \boldsymbol{\phi}_w, \boldsymbol{\psi}_f, \boldsymbol{\theta})$, denoted $\boldsymbol{\psi}_w(D^\tau)$ for brevity. There are three challenges with this approach: first, the dimensionality of the weights depends on the task ($\boldsymbol{\psi}_w^\tau$ is a matrix with a column for each class, see Figure 3) and thus the network must output parameters of different dimensionalities; second, the number of datapoints in $D^\tau$ will also depend on the task and so the network must be able to take inputs of variable cardinality; third, we would like the model to support continual learning. To handle the first two challenges we follow Gordon et al. [5]. First, each column of the weight matrix is generated independently from the context points from that class $\boldsymbol{\psi}_w^\tau = [\boldsymbol{\psi}_w\left(D_1^\tau\right), \quad \ldots, \quad \boldsymbol{\psi}_w\left(D_C^\tau\right)]$, an approach which scales to arbitrary numbers of classes. Second, we employ a permutation invariant architecture [18, 19] for $\boldsymbol{\psi}_w(\cdot)$ to handle the variable input cardinality (see Appendix E for details). Third, as permutation invariant architectures can be incrementally updated [20], continual learning is supported (as discussed in Section 5).

Intuitively, the classifier weights should be determined by the representation of the data points emerging from the adapted feature extractor. We therefore input the adapted feature representation of the data points into the network, rather than the raw data points (hence the dependency of $\boldsymbol{\psi}_w$ on $\boldsymbol{\psi}_f$ and $\boldsymbol{\theta}$). To summarize, $\boldsymbol{\psi}_w(\cdot)$ is a function *on sets* that accepts as input a set of *adapted* feature representations from $D_c^\tau$, and outputs the $c^{\text{th}}$ column of the linear classification matrix, i.e.,

$$\boldsymbol{\psi}_w\left(D_c^\tau; \boldsymbol{\phi}_w, \boldsymbol{\psi}_f, \boldsymbol{\theta}\right) = \boldsymbol{\psi}_w\left(\{f_{\boldsymbol{\theta}}\left(\boldsymbol{x}_m; \boldsymbol{\psi}_f\right) | \boldsymbol{x}_m \in D^\tau, \boldsymbol{y}_m = c\}; \boldsymbol{\phi}_w\right). \quad (2)$$

Here $\boldsymbol{\phi}_w$ are learnable parameters of $\boldsymbol{\psi}_w(\cdot)$. See Figure 3 for an illustration.

**Adaptation Network: Feature Extractor Parameters**. CNAPs represents the task-specific feature extractor parameters $\boldsymbol{\psi}_f^\tau$, comprising the parameters of the FiLM layers $\boldsymbol{\gamma}^\tau$ and $\boldsymbol{\beta}^\tau$ in our implementation, as a parameterized function of the context-set $D^\tau$. Thus, $\boldsymbol{\psi}_f(\cdot; \boldsymbol{\phi}_f, \boldsymbol{\theta})$ is a collection of functions (one for each FiLM layer) with parameters $\boldsymbol{\phi}_f$, many of which are shared across functions. We denote the function generating the parameters for the $i^{\text{th}}$ FiLM layer $\boldsymbol{\psi}_f^i(\cdot)$ for brevity.

Our experiments (Section 5) show that this mapping requires careful parameterization. We propose a novel parameterization that improves performance in complex settings with diverse datasets. Our implementation contains two components: a task-specific representation that provides context about the task to all layers of the feature extractor (denoted $\boldsymbol{z}_G^\tau$), and an auto-regressive component that provides information to deeper layers in the feature extractor concerning how shallower layers have adapted to the task (denoted $\boldsymbol{z}_{AR}^i$). The input to the $\boldsymbol{\psi}_f^i(\cdot)$ network is $\boldsymbol{z}_i = (\boldsymbol{z}_G^\tau, \boldsymbol{z}_{AR}^i)$. $\boldsymbol{z}_G^\tau$ is computed for every task $\tau$ by passing the inputs $\boldsymbol{x}_n^\tau$ through a global set encoder $g$ with parameters in $\boldsymbol{\phi}_f$.

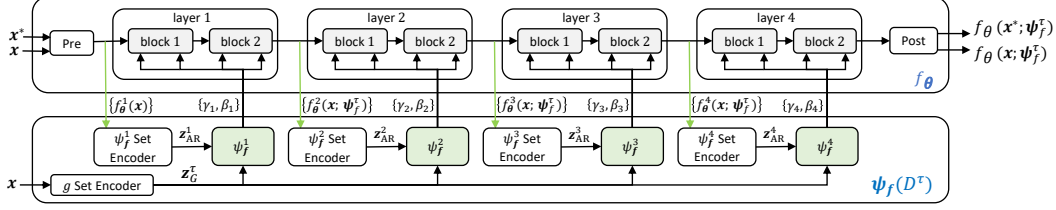

Figure 4: Implementation of the feature-extractor: an independently learned set encoder $g$ provides a fixed context that is concatenated to the (processed) activations of $\boldsymbol{x}$ from the previous ResNet block. The inputs $\boldsymbol{z}_i = (\boldsymbol{z}_{\mathrm{G}}^\tau, \boldsymbol{z}_{\mathrm{AR}}^i)$ are then fed to $\boldsymbol{\psi}_f^i(\cdot)$, which outputs the FiLM parameters for layer $i$. Green arrows correspond to propagation of auto-regressive representations. Note that the auto-regressive component $\boldsymbol{z}_{\mathrm{AR}}^i$ is computed by processing the *adapted* activations $\{f_\theta^i(\boldsymbol{x}; \boldsymbol{\psi}_f^\tau)\}$ of the previous convolutional block.

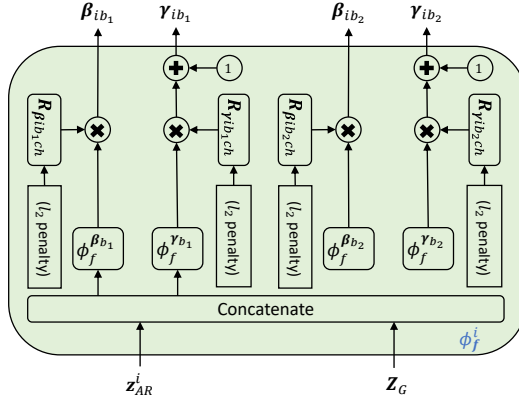

Figure 5: Adaptation network $\boldsymbol{\phi}_f$. $\boldsymbol{R}_{\gamma ib_j ch}$ and $\boldsymbol{R}_{\beta ib_j ch}$ denote a vector of regularization weights that are learned with an $l_2$ penalty.

To adapt the $l^{\mathrm{th}}$ layer in the feature extractor, it is useful for the system to have access to the representation of task-relevant inputs from layer $l-1$. While $\boldsymbol{z}_G$ could in principle encode how layer $l-1$ has adapted, we opt to provide this information directly to the adaptation network adapting layer $l$ by passing the adapted activations from layer $l-1$. The auto-regressive component $\boldsymbol{z}_{\mathrm{AR}}^i$ is computed by processing the *adapted* activations of the previous convolutional block with a layer-specific set encoder (except for the first residual block, whose auto-regressive component is given by the *un-adapted* initial pre-processing stage in the ResNet). Both the global and all layer-specific set-encoders are implemented as permutation invariant functions [18, 19] (see Appendix E for details). The full parameterization is illustrated in Figure 4, and the architecture of $\boldsymbol{\psi}_f^i(\cdot)$ networks is illustrated in Figure 5.

## 3 Model Training

The previous section has specified the model (see Figure 1b for a schematic). We now describe how to train the global classifier parameters $\boldsymbol{\theta}$ and the adaptation network parameters $\boldsymbol{\phi} = \{\boldsymbol{\phi}_f, \boldsymbol{\phi}_w\}$.

**Training the global classifier parameters $\boldsymbol{\theta}$.** A natural approach to training the model (originally employed by CNPs [13]) would be to maximize the likelihood of the training data jointly over $\boldsymbol{\theta}$ and $\boldsymbol{\phi}$. However, experiments (detailed in Appendix D.3) showed that it is crucially important to adopt a two stage process instead. In the first stage, $\boldsymbol{\theta}$ are trained on a large dataset (e.g., the training set of ImageNet [21, 6]) in a full-way classification procedure, mirroring standard pre-training. Second, $\boldsymbol{\theta}$ are fixed and $\boldsymbol{\phi}$ are trained using episodic training over all meta-training datasets in the multi-task setting. We hypothesize that two-stage training is important for two reasons: (i) during the second stage, $\boldsymbol{\phi}_f$ are trained to adapt $f_\theta(\cdot)$ to tasks $\tau$ by outputting $\boldsymbol{\psi}_f^\tau$. As $\boldsymbol{\theta}$ has far more capacity than $\boldsymbol{\psi}_f^\tau$, if they are trained in the context of all tasks, there is no need for $\boldsymbol{\psi}_f^\tau$ to adapt the feature extractor, resulting in little-to-no training signal for $\boldsymbol{\phi}_f$ and poor generalisation. (ii) Allowing $\boldsymbol{\theta}$ to adapt during

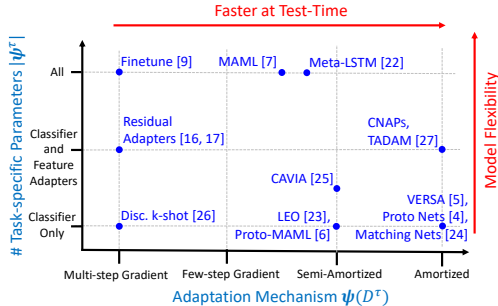

Figure 6: Model design space. The $y$-axis represents the number of task-specific parameters $|\psi^\tau|$. Increasing $|\psi^\tau|$ increases model flexibility, but also the propensity to over-fit. The $x$-axis represents the complexity of the mechanism used to adapt the task-specific parameters to training data $\psi(D^\tau)$. On the right are *amortized* approaches (i.e. using fixed functions). On the left is gradient-based adaptation. Mixed approaches lie between. Computational efficiency increases to the right. Flexibility increases to the left, but with it over-fitting and need for hand tuning.

the second phase violates the principle of "train as you test", i.e., when test tasks are encountered, $\theta$ will be fixed, so it is important to simulate this scenario during training. Finally, fixing $\theta$ during meta-training is desireable as it results in a dramatic decrease in training time.

**Training the adaptation network parameters $\phi$.** Following the work of Garnelo et al. [13], we train $\phi$ with maximum likelihood. An unbiased stochastic estimator of the log-likelihood is:

$$\hat{\mathcal{L}}(\phi) = \frac{1}{MT} \sum_{m,\tau} \log p\left(\boldsymbol{y}_m^{*\tau} | \boldsymbol{x}_m^{*\tau}, \boldsymbol{\psi}_\phi\left(D^\tau\right), \boldsymbol{\theta}\right), \tag{3}$$

where $\{\boldsymbol{y}_m^{*\tau}, \boldsymbol{x}_m^{*\tau}, D^\tau\} \sim \hat{P}$, with $\hat{P}$ representing the data distribution (e.g., sampling tasks and splitting them into disjoint context ($D^\tau$) and target data $\{(\boldsymbol{x}_m^{*\tau}, \boldsymbol{y}_m^{*\tau})\}_{m=1}^{M_t}$). Maximum likelihood training therefore naturally uses episodic context / target splits often used in meta-learning. In our experiments we use the protocol defined by Triantafillou et al. [6] and META-DATASET for this sampling procedure. Algorithm A.1 details computation of the stochastic estimator for a single task.

## 4 Related Work

Our work frames multi-task classification as directly modelling the predictive distribution $p(\boldsymbol{y}^* | \boldsymbol{x}^*, \boldsymbol{\psi}(D^\tau))$. The perspective allows previous work [7, 5, 15, 22, 16, 17, 23, 4, 6, 24, 9, 25, 26] to be organised in terms of i) the choice of the parameterization of the classifier (and in particular the nature of the local parameters), and ii) the function used to compute the local parameters from the training data. This space is illustrated in Figure 6, and further elaborated upon in Appendix B.

One of the inspirations for our work is conditional neural processes (CNPs) [13]. CNPs directly model the predictive distribution $p(\boldsymbol{y}^* | \boldsymbol{x}^*, \boldsymbol{\psi}(D^\tau))$ and train the parameters using maximum likelihood. Whereas previous work on CNPs has focused on homogeneous regression and classification datasets and fairly simple models, here we study multiple heterogeneous classification datasets and use a more complex model to handle this scenario. In particular, whereas the original CNP approach to classification required pre-specifying the number of classes in advance, CNAPs handles varying way classification tasks, which is required for e.g. the meta-dataset benchmark. Further, CNAPs employs a parameter-sharing hierarchy that parameterizes the feature extractor. This contrasts to the original CNP approach that shared all parameters across tasks, and use latent inputs to the decoder to adapt to new tasks. Finally, CNAPs employs a meta-training procedure geared towards *learning to adapt* to diverse tasks. Similarly, our work can be viewed as a deterministic limit of ML-PIP [5] which employs a distributional treatment of the local-parameters $\psi$.

A model with design choices closely related to CNAPs is TADAM [27]. TADAM employs a similar set of local parameters, allowing for adaptation of both the feature extractor and classification layer. However, it uses a far simpler adaptation network (lacking auto-regressive structure) and an expensive and ad-hoc training procedure. Moreover, TADAM was applied to simple few-shot learning benchmarks (e.g. CIFAR100 and mini-ImageNet) and sees little gain from feature extractor adaptation. In contrast, we see a large benefit from adapting the feature extractor. This may in part reflect the differences in the two models, but we observe that feature extractor adaptation has the largest impact when used to adapt to *different datasets* and that two stage training is required to see this. Further differences are our usage of the CNP framework and the flexible deployment of CNAPs to continual learning and active learning (see Section 5).

# 5   Experiments and Results

The experiments target three key questions: (i) Can CNAPs improve performance in multi-task few-shot learning? (ii) Does the use of an adaptation network benefit computational-efficiency and data-efficiency? (iii) Can CNAPs be deployed directly to complex learning scenarios like continual learning and active learning? The experiments use the following modelling choices (see Appendix E for full details). While CNAPs can utilize any feature extractor, a ResNet18 [14] is used throughout to enable fair comparison with Triantafillou et al. [6]. To ensure that each task is handled independently, batch normalization statistics [28] are learned (and fixed) during the pre-training phase for $\theta$. Actual batch statistics of the test data are never used during meta-training or testing.

**Few Shot Classification.**   The first experiment tackles a demanding few-shot classification challenge called META-DATASET [6]. META-DATASET is composed of ten (eight train, two test) image classification datasets. The challenge constructs few-shot learning tasks by drawing from the following distribution. First, one of the datasets is sampled uniformly; second, the "way" and "shot" are sampled randomly according to a fixed procedure; third, the classes and context / target instances are sampled. Where a hierarchical structure exists in the data (ILSVRC or OMNIGLOT), task-sampling respects the hierarchy. In the meta-test phase, the identity of the original dataset is not revealed and the tasks must be treated independently (i.e. no information can be transferred between them). Notably, the meta-training set comprises a disjoint and dissimilar set of classes from those used for meta-test. Full details are available in Appendix C.1 and [6].

Triantafillou et al. [6] consider two stage training: an initial stage that trains a feature extractor in a standard classification setting, and a meta-training stage of all parameters in an episodic regime. For the meta-training stage, they consider two settings: meta-training only on the META-DATASET version of ILSVRC, and on all meta-training data. We focus on the latter as CNAPs rely on training data from a variety of training tasks to learn to adapt, but provide results for the former in Appendix D.1. We pre-train $\theta$ on the meta-training set of the META-DATASET version of ILSVRC, and meta-train $\phi$ in an episodic fashion using all meta-training data. We compare CNAPs to models considered by Triantafillou et al. [6], including their proposed method (Proto-MAML) in Table 1. We meta-test CNAPs on three additional held-out datasets: MNIST [29], CIFAR10 [30], and CIFAR100 [30]. As an ablation study, we compare a version of CNAPs that does not make use of the auto-regressive component $z_{AR}$, and a version that uses no feature extractor adaptation. In our analysis of Table 1, we distinguish between two types of generalization: (i) unseen tasks (classes) in meta-training datasets, and (ii) unseen datasets.

**Unseen tasks:**   CNAPs achieve significant improvements over existing methods on seven of the eight datasets. The exception is the TEXTURES dataset, which has only seven test classes and accuracy is highly sensitive to the train / validation / test class split. The ablation study demonstrates that removing $z_{AR}$ from the feature extractor adaptation degrades accuracy in most cases, and that removing all feature extractor adaptation results in drastic reductions in accuracy.

**Unseen datasets:**   CNAPs-models outperform all competitive models with the exception of FINE-TUNE on the TRAFFIC SIGNS dataset. Removing $z_{AR}$ from the feature extractor decreases accuracy and removing the feature extractor adaptation entirely significantly impairs performance. The degradation is particularly pronounced when the held out dataset differs substantially from the dataset used to pretrain $\theta$, e.g. for MNIST.

Note that the superior results when using the auto-regressive component can not be attributed to increased network capacity alone. In Appendix D.4 we demonstrate that CNAPs yields superior classification accuracy when compared to parallel residual adapters [17] even though CNAPs requires significantly less network capacity in order to adapt the feature extractor to a given task.

**Additional results:**   Results when meta-training only on the META-DATASET version of ILSVRC are given in Table D.3. In Appendix D.2, we visualize the task encodings and parameters, demonstrating that the model is able to learn meaningful task and dataset level representations and parameterizations. The results support the hypothesis that learning to adapt key parts of the network is more robust and achieves significantly better performance than existing approaches.

| Dataset | Finetune | MatchingNet | ProtoNet | fo-MAML | Proto-MAML | CNAPs (no $\psi_f$) | CNAPs (no $z_{AR}$) | CNAPs |
|---|---|---|---|---|---|---|---|---|
| ILSVRC [21] | 43.1 ± 1.1 | 36.1 ± 1.0 | 44.5 ± 1.1 | 32.4 ± 1.0 | 47.9 ± 1.1 | 43.8 ± 1.0 | **51.3 ± 1.0** | **52.3 ± 1.0** |
| Omniglot [31] | 71.1 ± 1.4 | 78.3 ± 1.0 | 79.6 ± 1.1 | 71.9 ± 1.2 | 82.9 ± 0.9 | 60.1 ± 1.3 | **88.0 ± 0.7** | **88.4 ± 0.7** |
| Aircraft [32] | 72.0 ± 1.1 | 69.2 ± 1.0 | 71.1 ± 0.9 | 52.8 ± 0.9 | 74.2 ± 0.8 | 53.0 ± 0.9 | 76.8 ± 0.8 | **80.5 ± 0.6** |
| Birds [33] | 59.8 ± 1.2 | 56.4 ± 1.0 | 67.0 ± 1.0 | 47.2 ± 1.1 | 70.0 ± 1.0 | 55.7 ± 1.0 | **71.4 ± 0.9** | **72.2 ± 0.9** |
| Textures [34] | **69.1 ± 0.9** | 61.8 ± 0.7 | 65.2 ± 0.8 | 56.7 ± 0.7 | 67.9 ± 0.8 | 60.5 ± 0.8 | 62.5 ± 0.7 | 58.3 ± 0.7 |
| Quick Draw [35] | 47.0 ± 1.2 | 60.8 ± 1.0 | 64.9 ± 0.9 | 50.5 ± 1.2 | 66.6 ± 0.9 | 58.1 ± 1.0 | **71.9 ± 0.8** | 72.5 ± 0.8 |
| Fungi [36] | 38.2 ± 1.0 | 33.7 ± 1.0 | 40.3 ± 1.1 | 21.0 ± 1.0 | 42.0 ± 1.1 | 28.6 ± 0.9 | **46.0 ± 1.1** | **47.4 ± 1.0** |
| VGG Flower [37] | 85.3 ± 0.7 | 81.9 ± 0.7 | 86.9 ± 0.7 | 70.9 ± 1.0 | **88.5 ± 0.7** | 75.3 ± 0.7 | **89.2 ± 0.5** | 86.0 ± 0.5 |
| Traffic Signs [38] | **66.7 ± 1.2** | 55.6 ± 1.1 | 46.5 ± 1.0 | 34.2 ± 1.3 | 52.3 ± 1.1 | 55.0 ± 0.9 | 60.1 ± 0.9 | 60.2 ± 0.9 |
| MSCOCO [39] | 35.2 ± 1.1 | 28.8 ± 1.0 | 39.9 ± 1.1 | 24.1 ± 1.1 | **41.3 ± 1.0** | 41.2 ± 1.0 | 42.0 ± 1.0 | 42.6 ± 1.1 |
| MNIST [29] | | | | | | 76.0 ± 0.8 | 88.6 ± 0.5 | **92.7 ± 0.4** |
| CIFAR10 [30] | | | | | | **61.5 ± 0.7** | 60.0 ± 0.8 | **61.5 ± 0.7** |
| CIFAR100 [30] | | | | | | 44.8 ± 1.0 | 48.1 ± 1.0 | **50.1 ± 1.0** |

Table 1: Few-shot classification results on META-DATASET [6] using models trained on all training datasets. All figures are percentages and the ± sign indicates the 95% confidence interval over tasks. Bold text indicates the scores within the confidence interval of the highest score. Tasks from datasets below the dashed line were not used for training. Competing methods' results from [6].

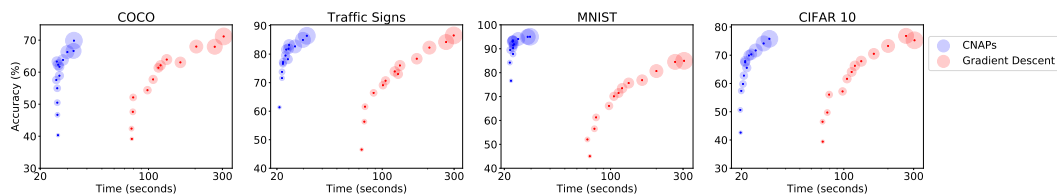

Figure 7: Comparing CNAPs to gradient based feature extractor adaptation: accuracy on 5-way classification tasks from withheld datasets as a function of processing time. Dot size reflects shot number (1 to 25 shots).

**FiLM Parameter Learning Performance: Speed-Accuracy Trade-off.** CNAPs generate FiLM layer parameters for each task $\tau$ at test time using the adaptation network $\psi_f(D^\tau)$. It is also possible to learn the FiLM parameters via gradient descent (see [16, 17]). Here we compare CNAPs to this approach. Figure 7 shows plots of 5-way classification accuracy versus time for four held out data sets as the number of shots was varied. For gradient descent, we used a fixed learning rate of 0.001 and took 25 steps for each point. The overall time required to produce the plot was 1274 and 7214 seconds for CNAPs and gradient approaches, respectively, on a NVIDIA Tesla P100-PCIE-16GB GPU. CNAPs is at least 5 times faster at test time than gradient-based optimization requiring only a single forward pass through the network while gradient based approaches require multiple forward and backward passes. Further, the accuracy achieved with adaptation networks is significantly higher for fewer shots as it protects against over-fitting. For large numbers of shots, gradient descent catches up, albeit slowly.

**Complex Learning Scenarios: Continual Learning.** In continual learning [40] new tasks appear over time and existing tasks may change. The goal is to adapt accordingly, but without retaining old data which is challenging for artificial systems. To demonstrate the the versatility CNAPs we show that, although it has not been explicitly trained for continual learning, we are able to apply the same model trained for the few-shot classification experiments (without the auto-regressive component) to standard continual learning benchmarks on held out datasets: Split MNIST [41] and Split CIFAR100 [42]. We modify the model to compute running averages for the representations of both $\psi_w^\tau$ and $\psi_f^\tau$ (see Appendix F for further details), in this way it performs incremental updates using the new data and the old model, and does not need to access old data. Figure 8 (left) shows the accumulated multi- and single-head [42] test accuracy averaged over 30 runs (further results and more detailed figures are in Appendix G). Figure 8 (right) shows average results at the final task comparing to SI [41], EWC [43], VCL [44], and Riemannian Walk [42].

Figure 8 demonstrates that CNAPs naturally resists catastrophic forgetting [43] and compares favourably to competing methods, despite the fact that it was not exposed to these datasets during training, observes orders of magnitude fewer examples, and was not trained explicitly to perform continual learning. CNAPs performs similarly to, or better than, the state-of-the-art Riemannian Walk method which departs from the pure continual learning setting by maintaining a small number of training samples across tasks. Conversely, CNAPs has the advantage of being exposed to a larger

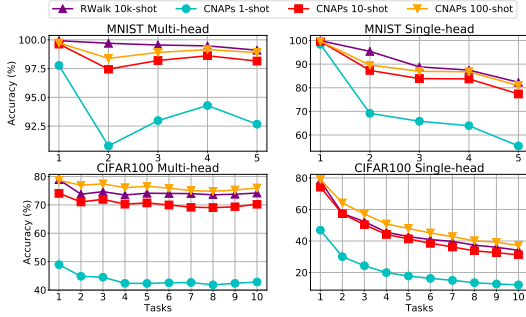

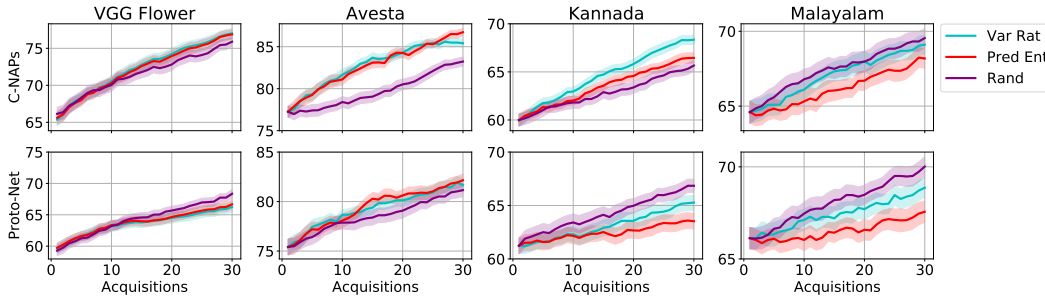

| Method | MNIST | | CIFAR100 | |
|---|---|---|---|---|
| | Multi | Single | Multi | Single |
| SI [41] | 99.3 | 57.6 | 73.2 | 22.8 |
| EWC [43] | 99.3 | 55.8 | 72.8 | 23.1 |
| VCL [44] | 98.5 ± 0.4 | - | - | - |
| RWalk [42] | 99.3 | 82.5 | 74.2 | 34.0 |
| CNAPs | 98.9 ± 0.2 | 80.9 ± 0.9 | 76.0 ± 0.5 | 37.2 ± 0.6 |

Figure 8: Continual learning classification results on Split MNIST and Split CIFAR100 using a model trained on all training datasets. (Left) The plots show accumulated accuracy averaged over 30 runs for both single- and multi-head scenarios. (Right) Average accuracy at final task computed over 30 experiments (all figures are percentages). Errors are one standard deviation. Additional results from [42, 45].

Figure 9: Accuracy vs active learning iterations for held-out classes / languages. (Top) CNAPs and (bottom) prototypical networks. Error shading is one standard error. CNAPs achieves better accuracy than prototypical networks and improvements over random acquisition, whereas prototypical networks do not.

range of datasets and can therefore leverage task transfer. We emphasize that this is not meant to be an "apples-to-apples" comparison, but rather, the goal is to demonstrate the out-of-the-box versatility and strong performance of CNAPs in new domains and learning scenarios.

**Complex Learning Scenarios: Active Learning**. Active learning [46, 47] requires accurate data-efficient learning that returns well-calibrated uncertainty estimates. Figure 9 compares the performance of CNAPs and prototypical networks using two standard active learning acquisition functions (variation ratios and predictive entropy [46]) against random acquisition on the FLOWERS dataset and three representative held-out languages from OMNIGLOT (performance on all languages is presented in Appendix H). Figure 9 and Appendix H show that CNAPs achieves higher accuracy on average than prototypical networks. Moreover, CNAPs achieves significant improvements over random acquisition, whereas prototypical networks do not. These tests indicates that CNAPs is more accurate and suggest that CNAPs has better calibrated uncertainty estimates than prototypical networks.

## 6 Conclusions

This paper has introduced CNAPs, an automatic, fast and flexible modelling approach for multi-task classification. We have demonstrated that CNAPs achieve state-of-the-art performance on the META-DATASET challenge, and can be deployed "out-of-the-box" to diverse learning scenarios such as continual and active learning where they are competitive with the state-of-the-art. Future avenues of research are to consider the exploration of the design space by introducing gradients and function approximation to the adaptation mechanisms, as well as generalizing the approach to distributional extensions of CNAPs [48, 49].

## Acknowledgments

The authors would like to thank Ambrish Rawat for helpful discussions and David Duvenaud, Wessel Bruinsma, Will Tebbutt Adrià Garriga Alonso, Eric Nalisnick, and Lyndon White for the insightful comments and feedback. Richard E. Turner is supported by Google, Amazon, Improbable and EPSRC grants EP/M0269571 and EP/L000776/1.

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
