[Supplementary Material]

# Supplementary Material for Fast and Flexible Multi-Task Classification Using Conditional Neural Adaptive Processes

**James Requeima**[*]
University of Cambridge
Invenia Labs
jrr41@cam.ac.uk

**Jonathan Gordon**[*]
University of Cambridge
jg801@cam.ac.uk

**John Bronskill**[*]
University of Cambridge
jfb54@cam.ac.uk

**Sebastian Nowozin**
Google Research Berlin
nowozin@google.com

**Richard E. Turner**
University of Cambridge
Microsoft Research
ret26@cam.ac.uk

## A  Algorithm for Constructing Stochastic Estimator

An algorithm for constructing the stochastic training objective $\hat{\mathcal{L}}(\boldsymbol{\phi}; \tau)$ for a single task $\tau$ is given in Algorithm A.1. $\text{CAT}(\cdot; \boldsymbol{\pi})$ denotes a the likelihood of a categorical distribution with parameter vector $\boldsymbol{\pi}$. This algorithm can be used on a batch of tasks to construct an unbiased estimator for the auto-regressive likelihood of the task outputs.

---

**Algorithm A.1** Stochastic Objective Estimator for Meta-Training.

1: **procedure** META-TRAINING($\{\boldsymbol{x}_m^*, \boldsymbol{y}_m^*\}_{m=1}^M, D^\tau, \boldsymbol{\theta}, \boldsymbol{\phi}$)
2:     $\boldsymbol{\psi}_f^\tau \leftarrow \boldsymbol{\psi}_f(\{f_{\boldsymbol{\theta}}(\boldsymbol{x}_n)|\boldsymbol{x} \in D^\tau\}; \boldsymbol{\phi}_f)$
3:     $\boldsymbol{\psi}_c^\tau \leftarrow \boldsymbol{\psi}_w(\{f_{\boldsymbol{\theta}}(\boldsymbol{x}_n; \boldsymbol{\psi}_f)|\boldsymbol{x} \in D^\tau, \boldsymbol{y}_n = c\}; \boldsymbol{\phi}_w)$   $\forall c \in C^\tau$
4:     **for** $m \in 1, ..., M$ **do**
5:         $\boldsymbol{\pi}_m \leftarrow f_{\boldsymbol{\theta}}(\boldsymbol{x}_m^*; \boldsymbol{\psi}_f^\tau)^T \boldsymbol{\psi}_w^\tau$
6:         $\log p(\boldsymbol{y}_m^*|\boldsymbol{\pi}_m) \leftarrow \log \text{CAT}(\boldsymbol{y}_m^*; \boldsymbol{\pi}_m)$
7:     **end for**
8:     **return** $\hat{\mathcal{L}}(\boldsymbol{\phi}; \tau) \leftarrow \frac{1}{M} \sum_M \log p(\boldsymbol{y}_m^*|\boldsymbol{\pi}_m)$

9: **end procedure**

---

## B  Additional Related Work Details

**The choice of task-specific parameters $\boldsymbol{\psi}^\tau$.**   Clearly, any approach to multi-task classification must adapt, at the very least, the top-level classifier layer of the model. A number of successful models have proposed doing just this with e.g., neighbourhood-based approaches [1], variational inference [2], or inference networks [3]. On the other end of the spectrum are models that adapt *all* the parameters of the classifier, e.g., [4, 5, 6]. The trade-off here is clear: as more parameters are adapted, the resulting model is more flexible, but also slow and prone to over-fitting. For this reason we modulate a small portion of the network parameters, following recent work on multi-task learning [7, 8, 9].

---

[*]Authors contributed equally

We argue that just adapting the linear classification layer is sufficient when the task distribution is not diverse, as in the standard benchmarks used for few-shot classification (OMNIGLOT [10] and *mini*-imageNet [11]). However, when faced with a diverse set of tasks, such as that introduced recently by Triantafillou et al. [12], it is important to adapt the feature extractor on a per-task basis as well.

**The adaptation mechanism $\psi_\phi (D^\tau)$.** Adaptation varies in the literature from performing full gradient descent learning with $D^\tau$ [13] to relying on simple operations such as taking the mean of class-specific feature representations [1, 14]. Recent work has focused on reducing the number of required gradient steps by learning a global initialization [4, 5] or additional parameters of the optimization procedure [11]. Gradient-based procedures have the benefit of being flexible, but are computationally demanding, and prone to over-fitting in the low-data regime. Another line of work has focused on learning neural networks to output the values of $\psi$, which we denote *amortization* [3]. Amortization greatly reduces the cost of adaptation and enables sharing of global parameters, but may suffer from the amortization gap [15] (i.e., underfitting), particularly in the large data regime. Recent work has proposed using semi-amortized inference [12, 16], but have done so while only adapting the classification layer parameters.

## C Experimentation Details

All experiments were implemented in PyTorch [17] and executed either on NVIDIA Tesla P100-PCIE-16GB or Tesla V100-SXM2-16GB GPUs. The full CNAPs model runs in a distributed fashion across 2 GPUs and takes approximately one and a half days to complete episodic training and testing.

### C.1 META-DATASET Training and Evaluation Procedure

#### C.1.1 Feature Extractor Weights $\theta$ Pretraining

We first reduce the size of the images in the ImageNet ILSVRC-2012 dataset [18] to $84 \times 84$ pixels. Some images in the ImageNet ILSVRC-2012 dataset are duplicates of images in other datasets included in META-DATASET, so these were removed. We then split the 1000 training classes of the ImageNet ILSVRC-2012 dataset into training, validation, and test sets according to the criteria detailed in [12]. The test set consists of the 130 leaf-node subclasses of the "device" synset node, the validation set consists of the the 158 leaf-node subclasses of the "carnivore" synset node, and the training set consists of the remaining 712 leaf-node classes. We then pretrain a feature extractor with parameters $\theta$ based on a modified ResNet-18 [19] architecture on the above 712 training classes. The ResNet-18 architecture is detailed in Table E.8. Compared to a standard ResNet-18, we reduced the initial convolution kernel size from 7 to 5 and eliminated the initial max-pool step. These changes were made to accommodate the reduced size of the imagenet training images. We train for 125 epochs using stochastic gradient descent with momentum of 0.9, weight decay equal to 0.0001, a batch size of 256, and an initial learning rate of 0.1 that decreases by a factor of 10 every 25 epochs. During pretraining, the training dataset was augmented with random crops, random horizontal flips, and random color jitter. The top-1 accuracy after pretraining was 63.9%. For all subsequent training and evaluation steps, the ResNet-18 weights were frozen.The dimensionality of the feature extractor output is $d_f = 512$. The hyper-parameters used were derived from the PyTorch [17] ResNet training tutorial. The only tuning that was performed was on the number of epochs used for training and the interval at which the learning rate was decreased. For the number of epochs, we tried both 90 and 125 epochs and selected 125, which resulted in slightly higher accuracy. We also found that dropping the learning rate at an interval of 25 versus 30 epochs resulted in slightly higher accuracy.

#### C.1.2 Episodic Training of $\phi$

Next we train the functions that generate the parameters $\psi_f^\tau$, $\psi_w^\tau$ for the feature extractor adapters and the linear classifier, respectively. We train two variants of CNAPs (on ImageNet ILSVRC-2012 only and all datasets - see Table C.2). We generate training and validation episodes using the reader from [20]. We train in an end-to-end fashion for 110,000 episodes with the Adam [21] optimizer, using a batch size of 16 episodes, and a fixed learning rate of 0.0005. We validate using 200 episodes per validation dataset. Note that when training on ILSVRC only, we validate on ILSVRC only, however, when training on all datasets, we validate on all datasets that have validation data (see Table C.2) and

consider a model to be better if more than half of the datasets have a higher classification accuracy than the current best model. No data augmentation was employed during the training of $\phi$. Note that while training $\phi$ the feature extractor $f_{\theta}(\cdot)$ is in 'eval' mode (i.e. it will use the fixed batch normalization statistics learned during pretraining the feature extractor weights $\theta$ with a moving average). No batch normalization is used in any of the functions generating the $\psi^{\tau}$ parameters, with the exception of the set encoder $g$ (that generates the global task representation $z_{G}^{\tau}$). Note that the target points are never passed through the set encoder $g$. Again, very little hyper-parameter tuning was performed. No grid search or other hyper-parameter search was used. For learning rate we tried both 0.0001 and 0.0005, and selected the latter. We experimented with the number of training episodes in the range of 80,000 to 140,000, with 110,000 episodes generally yielding the best results. We also tried lowering the batch size to 8, but that led to decreased accuracy.

### C.1.3 Evaluation

We generate test episodes using the reader from [20]. We test all models with 600 episodes each on all test datasets. The classification accuracy is averaged over the episodes and a 95% confidence interval is computed. We compare the best validation and fully trained models in terms of accuracy and use the best of the two. Note that during evaluation, the feature extractor $f_{\theta}(\cdot)$ is also in 'eval' mode.

| ImageNet ILSVRC-2012 | | | All Datasets | | |
|---|---|---|---|---|---|
| **Train** | **Validation** | **Test** | **Train** | **Validation** | **Test** |
| ILSVRC [18] | ILSVRC [18] | ILSVRC [18] | ILSVRC [18] | ILSVRC [18] | ILSVRC [18] |
| | | Omniglot [10] | Omniglot [10] | Omniglot [10] | Omniglot [10] |
| | | Aircraft [22] | Aircraft [22] | Aircraft [22] | Aircraft [22] |
| | | Birds [23] | Birds [23] | Birds [23] | Birds [23] |
| | | Textures [24] | Textures [24] | Textures [24] | Textures [24] |
| | | Quick Draw [25] | Quick Draw [25] | Quick Draw [25] | Quick Draw [25] |
| | | Fungi [26] | Fungi [26] | Fungi [26] | Fungi [26] |
| | | VGG Flower [27] | VGG Flower [27] | VGG Flower [27] | VGG Flower [27] |
| | | MSCOCO [28] | | MSCOCO [28] | MSCOCO [28] |
| | | Traffic Signs [29] | | | Traffic Signs [29] |
| | | MNIST [30] | | | MNIST [30] |
| | | CIFAR10 [31] | | | CIFAR10 [31] |
| | | CIFAR100 [31] | | | CIFAR100 [31] |

Table C.2: Datasets used to train, validate, and test models.

## D  Additional Few-Shot Classification Results

### D.1  Few-Shot Classification Results When Training on ILSVRC-2012 only

Table D.3 shows few-shot classification results on META-DATASET when trained on ILSVRC-2012 only. We emphasize that this scenario does not capture the key focus of our work, and that these results are provided mainly for completeness and compatibility with the work of Triantafillou et al. [12]. In particular, our method relies on training the parameters $\phi$ to adapt the conditional predictive distribution to new datasets. In this setting, the model is never presented with data that has not been used to pre-train $\theta$, and therefore cannot learn to appropriately adapt the network to new datasets. Despite this, CNAPs demonstrate competitive results with the methods evaluated by Triantafillou et al. [12] even in this scenario.

### D.2  Feature Extractor Parameter Learning

Figure D.10 shows t-SNE [32] plots that visualize the output of the set encoder $z_{G}$ and the FiLM layer parameters following the first and last convolutional layers of the feature extractor at test time. Even with unseen test data, the set encoder has learned to clearly separate examples arising from diverse datasets. The FiLM generators learn to generate feature extractor adaptation parameters unique to each dataset. The only significant overlap in the FiLM parameter plots is between CIFAR10 and CIFAR100 datasets which are closely related.

| Dataset | Finetune | MatchingNet | ProtoNet | fo-MAML | Proto-MAML | CNAPs |
|---|---|---|---|---|---|---|
| ILSVRC [18] | 45.8±1.1 | 45.0±1.1 | **50.5±1.1** | 36.1±1.0 | **51.0±1.1** | **50.6±1.1** |
| Omniglot [10] | **60.9±1.6** | 52.3±1.3 | 60.0±1.4 | 38.7±1.4 | **63.0±1.4** | 45.2±1.4 |
| Aircraft [22] | **68.7±1.3** | 49.0±0.9 | 53.1±1.0 | 34.5±0.9 | 55.3±1.0 | 36.0±0.8 |
| Birds [23] | 57.3±1.3 | 62.2±1.0 | **68.8±1.0** | 49.1±1.2 | **66.9±1.0** | 60.7±0.9 |
| Textures [24] | **69.1±0.9** | 64.2±0.9 | 66.6±0.8 | 56.5±0.8 | **67.8±0.8** | 67.5±0.7 |
| Quick Draw [25] | 42.6±1.2 | 42.9±1.1 | 49.0±1.1 | 27.2±1.2 | **53.7±1.1** | 42.3±1.0 |
| Fungi [26] | **38.2±1.0** | 34.0±1.0 | **39.7±1.1** | 23.5±1.0 | **38.0±1.1** | 30.1±0.9 |
| VGG Flower [27] | 85.5±0.7 | 80.1±0.7 | **85.3±0.8** | 66.4±1.0 | **86.9±0.8** | 70.7±0.7 |
| Traffic Signs [29] | 66.8±1.3 | 47.8±1.1 | 47.1±1.1 | 33.2±1.3 | 51.2±1.1 | 53.3±0.9 |
| MSCOCO [28] | 34.9±1.0 | 35.0±1.0 | 41.0±1.1 | 27.5±1.1 | **43.4±1.1** | 45.2±1.1 |
| MNIST [30] | | | | | | 70.4±0.8 |
| CIFAR10 [31] | | | | | | 65.2±0.8 |
| CIFAR100 [31] | | | | | | 53.6±1.0 |

Table D.3: Few-shot classification results on META-DATASET [12] using models trained on ILSVRC-2012 only. All figures are percentages and the ± sign indicates the 95% confidence interval. Bold text indicates the highest scores that overlap in their confidence intervals. Results from competitive methods from [12]

Figure D.10: t-SNE plots of the output of the set encoder $z_G$ and the FiLM layer parameters at the start $(\beta_{1b1}, \gamma_{1b1})$ and end $(\beta_{4b2}, \gamma_{4b2})$ of the feature extraction process at test time.

## D.3 Joint Training of $\theta$ and $\phi$

Our experiments in jointly training $\theta$ and $\phi$ show that the two-stage training procedure proposed in Section 3 is crucially important. In particular, we found that joint training diverged in almost all cases we attempted. We were only able to train jointly in two circumstances: (i) Using batch normalization in "train" mode for both context *and* target sets. We stress that this implies computing the batch statistics at test time, and using those to normalize the batches. This is in contrast to the methodology we propose in the main text: only using batch normalization in "eval" mode, which enforces that no information is transferred across tasks or datasets. (ii) "Warm-start" the training procedure with batch normalization in "train" mode, and after a number of epochs (we use 50 for the results shown below), switch to proper usage of batch normalization. All other training procedures we attempted diverged.

Table D.4 details the results of our study on training procedures. The results demonstrate that the two-stage greatly improves performance of the model, even compared to using batch normalization in "train mode", which gives the model an unfair advantage over our standard model.

## D.4 Comparison Between CNAPs and Parallel Residual Adapters [8]

CNAPs adds FiLM layers [9] in series with each convolutional layer to adapt the feature extractor to a particular task while parallel residual adapters from Rebuffi et al. [8] adds $1 \times 1$ convolutions in parallel with each convolution layer to do the same. However, if the number of feature channels is $C$, then the number of parameters required for each convolutional layer in the feature extractor is $2C$ for CNAPs and $C^2$ for parallel residual adapters. Hence, parallel residual adapters have $C/2$ times the capacity compared to FiLM layers. Despite this advantage, CNAPs achieves superior results as can be seen in Table D.5.

| Dataset | Joint Training (warmstart BN) | Joint Training (BN train mode) | Two-Stage Training (BN test mode) |
|---|---|---|---|
| ILSVRC [18] | 17.3±0.7 | 41.6±1.0 | 49.5±1.0 |
| Omniglot [10] | 74.9±1.0 | 80.8±0.9 | 89.7±0.5 |
| Aircraft [22] | 51.4±0.8 | 70.5±0.7 | 87.2±0.5 |
| Birds [23] | 44.1±1.0 | 48.3±1.0 | 76.7±0.9 |
| Textures [24] | 49.1±0.7 | 73.5±0.6 | 83.0±0.6 |
| Quick Draw [25] | 46.6±1.0 | 71.5±0.8 | 72.3±0.8 |
| Fungi [26] | 20.4±0.9 | 43.1±1.1 | 50.5±1.1 |
| VGG Flower [27] | 66.6±0.8 | 71.0±0.7 | 92.5±0.4 |
| Traffic Signs [29] | 21.2±0.8 | 40.4±1.1 | 48.4±1.1 |
| MSCOCO [28] | 18.8±0.7 | 37.1±1.0 | 39.7±0.9 |

Table D.4: Few-shot classification results on META-DATASET [12] comparing joint training for $\theta$ and $\phi$ (columns 2 and 3) to two-stage training (column 4). All figures are percentages and the $\pm$ sign indicates the 95% confidence interval. Bold text indicates the highest scores that overlap in their confidence intervals.

| Dataset | Parallel Residual Adapter | CNAPs |
|---|---|---|
| ILSVRC [18] | **51.2 ± 1.0** | **52.3 ± 1.0** |
| Omniglot [10] | **87.3 ± 0.7** | **88.4 ± 0.7** |
| Aircraft [22] | 78.3 ± 0.7 | **80.5 ± 0.6** |
| Birds [23] | 67.8 ± 0.9 | **72.2 ± 0.9** |
| Textures [24] | 55.5 ± 0.7 | **58.3 ± 0.7** |
| Quick Draw [25] | 70.9 ± 0.7 | **72.5 ± 0.8** |
| Fungi [26] | 44.6 ± 1.1 | **47.4 ± 1.0** |
| VGG Flower [27] | 81.7 ± 0.7 | **86.0 ± 0.5** |
| Traffic Signs [29] | 57.2 ± 0.9 | **60.2 ± 0.9** |
| MSCOCO [28] | **43.7 ± 1.0** | 42.6 ± 1.1 |
| MNIST [30] | 91.1 ± 0.4 | **92.7 ± 0.4** |
| CIFAR10 [31] | **64.5 ± 0.8** | 61.5 ± 0.7 |
| CIFAR100 [31] | **50.4 ± 0.9** | **50.1 ± 1.0** |

Table D.5: Few-shot classification results on META-DATASET [12] using models trained on all training datasets for Parallel Residual Adapters [8] and CNAPs. All figures are percentages and the $\pm$ sign indicates the 95% confidence interval over tasks. Bold text indicates the scores within the confidence interval of the highest score. Tasks from datasets below the dashed line were not used for training.

# E  Network Architecture Details

## E.1  ResNet18 Architecture details

Throughout our experiments in Section 5, we use a ResNet18 [19] as our feature extractor, the parameters of which we denote $\theta$. Table E.6 and Table E.7 detail the architectures of the basic block (left) and basic scaling block (right) that are the fundamental components of the ResNet that we employ. Table E.8 details how these blocks are composed to generate the overall feature extractor network. We use the implementation that is provided by the PyTorch [17][3], though we adapt the code to enable the use of FiLM layers.

## E.2  Adaptation Network Architecture Details

In this section, we provide the details of the architectures used for our adaptation networks. Table E.9 details the architecture of the set encoder $g : D^\tau \mapsto z_\text{G}$ that maps context sets to global representations.

Table E.10 details the architecture used in the auto-regressive parameterization of $z_\text{AR}$. In our experiments, there is one such network for every block in the ResNet18 (detailed in Table E.8). These

Table E.6: ResNet-18 basic block $b$.

| Layers |
| --- |
| Input |
| Conv2d ($3 \times 3$, stride 1, pad 1) |
| BatchNorm |
| FiLM ($\boldsymbol{\gamma}_{b,1}, \boldsymbol{\beta}_{b,1}$) |
| ReLU |
| Conv2d ($3 \times 3$, stride 1, pad 1) |
| BatchNorm |
| FiLM ($\boldsymbol{\gamma}_{b,2}, \boldsymbol{\beta}_{b,2}$) |
| Sum with Input |
| ReLU |

Table E.7: ResNet-18 basic scaling block $b$.

| Layers |
| --- |
| Input |
| Conv2d ($3 \times 3$, stride 2, pad 1) |
| BatchNorm |
| FiLM ($\boldsymbol{\gamma}_{b,1}, \boldsymbol{\beta}_{b,1}$) |
| ReLU |
| Conv2d ($3 \times 3$, stride 1, pad 1) |
| BatchNorm |
| FiLM ($\boldsymbol{\gamma}_{b,2}, \boldsymbol{\beta}_{b,2}$) |
| Downsample Input by factor of 2 |
| Sum with Downsampled Input |
| ReLU |

**ResNet-18 Feature Extractor ($\theta$) with FiLM Layers:** $\boldsymbol{x} \rightarrow f_{\boldsymbol{\theta}}(\boldsymbol{x}; \boldsymbol{\psi}_f^\tau)$, $\boldsymbol{x}^* \rightarrow f_{\boldsymbol{\theta}}(\boldsymbol{x}^*; \boldsymbol{\psi}_f^\tau)$

| Stage | Output size | Layers |
| --- | --- | --- |
| Input | $84 \times 84 \times 3$ | Input image |
| Pre-processing | $41 \times 41 \times 64$ | Conv2d ($5 \times 5$, stride 2, pad 1, BatchNorm, ReLU) |
| Layer 1 | $41 \times 41 \times 64$ | Basic Block $\times 2$ |
| Layer 2 | $21 \times 21 \times 128$ | Basic Block, Basic Scaling Block |
| Layer 3 | $11 \times 11 \times 256$ | Basic Block, Basic Scaling Block |
| Layer 4 | $6 \times 6 \times 512$ | Basic Block, Basic Scaling Block |
| Post-Processing | 512 | AvgPool, Flatten |

Table E.8: ResNet-18 feature extractor network.

networks accept as input the set of activations from the previous block, and map them (through the permutation invariant structure) to a vector representation of the output of the layer. The representation $\boldsymbol{z}_i = (\boldsymbol{z}_{\mathrm{G}}, \boldsymbol{z}_{\mathrm{AR}})$ is then generated by concatenating the global and auto-regressive representations, and fed into the adaptation network that provides the FiLM layer parameters for the next layer. This network is detailed in Table E.11, and illustrated in Figure 5. Note that, as depicted in Figure 5, each layer has four networks with architectures as detailed in Table E.11, one for each $\boldsymbol{\gamma}$ and $\boldsymbol{\beta}$, for each convolutional layer in the block.

### E.3 Linear Classifier Adaptation Network

Finally, in this section we give the details for the linear classifer $\boldsymbol{\psi}_w^\tau$, and the adaptation network that provides these task-specific parameters $\boldsymbol{\psi}_w(\cdot)$. The adaptation network accepts a class-specific representation that is generated by applying a mean-pooling operation to the adapted feature activations of each instance associated with the class in the context set: $\boldsymbol{z}_c^\tau = \frac{1}{N_c^\tau} \sum_{\boldsymbol{x} \in D_c^\tau} f_{\boldsymbol{\theta}}(\boldsymbol{x}; \boldsymbol{\psi}_f^\tau)$, where $N_c^\tau$

denotes the number of context instances associated with class $c$ in task $\tau$. $\boldsymbol{\psi}_w$ is comprised of two separate networks (one for the weights $\boldsymbol{\psi}_w$ and one for the biases $\boldsymbol{\psi}_b$) detailed in Table E.12 and Table E.13. The resulting weights and biases (for each class in task $\tau$) can then be used as a linear classification layer, as detailed in Table E.14.

**Set Encoder** ($g$): $\boldsymbol{x} \to \boldsymbol{z}_G^\tau$

| Output size | Layers |
|---|---|
| $84 \times 84 \times 3$ | Input image |
| $42 \times 42 \times 64$ | Conv2d ($3 \times 3$, stride 1, pad 1, ReLU), MaxPool ($2 \times 2$, stride 2) |
| $21 \times 21 \times 64$ | Conv2d ($3 \times 3$, stride 1, pad 1, ReLU), MaxPool ($2 \times 2$, stride 2) |
| $10 \times 10 \times 64$ | Conv2d ($3 \times 3$, stride 1, pad 1, ReLU), MaxPool ($2 \times 2$, stride 2) |
| $5 \times 5 \times 64$ | Conv2d ($3 \times 3$, stride 1, pad 1, ReLU), MaxPool ($2 \times 2$, stride 2) |
| $2 \times 2 \times 64$ | Conv2d ($3 \times 3$, stride 1, pad 1, ReLU), MaxPool ($2 \times 2$, stride 2) |
| $64$ | AdaptiveAvgPool2d |

Table E.9: Set encoder $g$.

**Set Encoder** ($\phi_f$): $\{f_{\boldsymbol{\theta}}^{l_i}(x; \boldsymbol{\psi}_f^\tau)\} \to \boldsymbol{z}_{\text{AR}}^i$

| Output size | Layers |
|---|---|
| $l_i$ channels $\times l_i$ channel size | Input $\{f_{\boldsymbol{\theta}}^{l_i}(x; \boldsymbol{\psi}_f^\tau)\}$ |
| $l_i$ channels $\times l_i$ channel size | AvgPool, Flatten |
| $l_i$ channels | fully connected, ReLU |
| $l_i$ channels | $2 \times$ fully connected with residual skip connection, ReLU |
| $l_i$ channels | fully connected with residual skip connection |
| $l_i$ channels | mean pooling over instances |
| $l_i$ channels | Input from mean pooling |
| $l_i$ channels | fully connected, ReLU |

Table E.10: Network of set encoder $\phi_f$.

**Network** ($\phi_f$): $(\boldsymbol{z}_{\text{G}}, \boldsymbol{z}_{\text{AR}}) \to (\boldsymbol{\gamma}, \boldsymbol{\beta})$

| Output size | Layers |
|---|---|
| $64 + l_i$ channels | Input from Concatenate |
| $l_i$ channels | fully connected, ReLU |
| $l_i$ channels | $2 \times$ fully connected with residual skip connection, ReLU |
| $l_i$ channels | fully connected with residual skip connection |

Table E.11: Network $\phi_f$.

Table E.12: Network $\phi_w$.

**Network** ($\phi_w$):
$\boldsymbol{z}_c \to \boldsymbol{\psi}_{w,w}$

| Output size | Layers |
|---|---|
| 512 | Input from mean pooling |
| 512 | $2 \times$ fully connected, ELU |
| 512 | fully connected |
| 512 | Sum with Input |

Table E.13: Network $\phi_b$.

**Network** ($\phi_b$):
$\boldsymbol{z}_c \to \boldsymbol{\psi}_{w,b}$

| Output size | Layers |
|---|---|
| 512 | Input from mean pooling |
| 512 | $2 \times$ fully connected, ELU |
| 1 | fully connected |

**Linear Classifier** ($\boldsymbol{\psi}_w$): $f_{\boldsymbol{\theta}}(\boldsymbol{x}^*; \boldsymbol{\psi}_f^\tau) \to p(\boldsymbol{y}^*|\boldsymbol{x}^*, \boldsymbol{\psi}^\tau(D^\tau), \boldsymbol{\theta})$

| Output size | Layers |
|---|---|
| 512 | Input features $f_{\boldsymbol{\theta}}(\boldsymbol{x}^*; \boldsymbol{\psi}_f^\tau)$ |
| $512 \times C^\tau$ | Input weights $w$ |
| $512 \times 1$ | Input biases $b$ |
| $C^\tau$ | fully connected |
| $C^\tau$ | softmax |

Table E.14: Linear classifier network.

# F  Continual Learning Implementation Details

As noted in Sections 2 and 5, our model can be applied to continual learning with one small modification: we store a compact representation of our training data that can be updated at each step of the continual learning procedure. Notice that Figure 3 indicates that the functional representation of our linear classification layer $\boldsymbol{\psi}_w^\tau(\cdot)$ contains a mean pooling layer that combines the per-class output of our feature extractor $\{f_{\boldsymbol{\theta}}\left(\boldsymbol{x}_m^\tau;\boldsymbol{\psi}_f\right)|\boldsymbol{x}_m^\tau \in D^\tau, \boldsymbol{y}_m^\tau = c\}$. The result of this pooling,

$$\boldsymbol{z}_c = \frac{1}{M}\sum f_{\boldsymbol{\theta}}\left(\boldsymbol{x}_m^\tau;\boldsymbol{\psi}_f\right) \tag{F.4}$$

where $M = |\{f_{\boldsymbol{\theta}}\left(\boldsymbol{x}_m^\tau;\boldsymbol{\psi}_f\right)|\boldsymbol{x}_m^\tau \in D^\tau, \boldsymbol{y}_m^\tau = c\}|$, is supplied as input to the network $\boldsymbol{\psi}_w(\cdot)$. This network yields the class conditional parameters of the linear classifier $\boldsymbol{\psi}_w^\tau$, resulting in (along with the feature extractor parameters $\boldsymbol{\psi}_f^\tau$) the full paramterization of $\boldsymbol{\psi}^\tau$. We store $\boldsymbol{z}_c$ as the training dataset representation for, class $c$.

If at any point in our continual learning procedure we observe new training data for class $c$ we can update our representation for class $c$ by computing $\boldsymbol{z}_c' = \frac{1}{M}\sum f_{\boldsymbol{\theta}}\left(\boldsymbol{x}_m^\tau{}';\boldsymbol{\psi}_f\right)$ the pooled average resulting from $M$ new training examples $\boldsymbol{x}_m^\tau{}'$ for class $c$. We then update $\boldsymbol{z}_c$ with the weighted average: $\boldsymbol{z}_c \leftarrow \frac{M\boldsymbol{z}_c' + N\boldsymbol{z}_c}{M+N}$. At prediction time, we supply $\boldsymbol{z}_c$ to $\boldsymbol{\psi}_w(\cdot)$ to produce classification parameters for class $c$.

Similar to the input to $\boldsymbol{\psi}_w^\tau(\cdot)$, the input to $\boldsymbol{\psi}_f^\tau(\cdot)$ also contains a mean-pooled representation, this time of the entire training dataset $\boldsymbol{z}_G^\tau$. This representation is also stored and updated in the same way.

One issue with our procedure is that it is not completely invariant to the order in which we observe the sequence of training data during our continual learning procedure. The feature extractor adaptation parameters are only conditioned on the most recent training data, meaning that if data from class $c$ is not present in the most recent training data, $z_c$ was generated using "old" feature extractor adaptation parameters (from a previous time step). This creates a potential disconnect between the classification parameters from previous time steps and the feature extractor output. Fortunately, in our experiment we noticed little within dataset variance for the adaptation parameters. Since all of our experiments on continual learning were within a single dataset, this did not seem to be an issue as CNAPs were able to achieved good performance. However, for continual learning experiments that contain multiple datasets, we anticipate that this issue will need to be addressed.

# G  Additional Continual Learning Results

In Section 5 we provided results for continual learning experiments with Split MNIST [33] and Split CIFAR100 [34]. The results showed the average performance as more tasks were observed for the single and multi head settings. Here, we provide more complete results, detailing the performance through "time" at the task level. Figure G.11 details the performance of CNAPs (with varying

Figure G.11: Continual learning results on Split MNIST. Top row is multi-head, bottom row is single-head.

number of observed examples) and Riemannian Walk (RWalk) [34] on the five tasks of Split MNIST

through time. Note that RWalk makes explicit use of training data from previous time steps when new data is observed, while CNAPs do not.

Figure G.11 implies that CNAPs is competitive with RWalk in this scenario, despite seeing far less data per task, and not using old data to retrain the model at every time-step. Further, we see that CNAPs is naturally resistant to forgetting, as it uses internal task representations to maintain important information about tasks seen at previous time-steps.

Figure G.12 demonstrates that CNAPs maintains similar results when scaling up to considerably more difficult datasets such as CIFAR100. Here too, CNAPs has not been trained on this dataset, yet demonstrates performance comparable to (and even better than) RWalk, a method explicitly trained for this task that makes use of samples from previous tasks at each time step.

Figure G.12: Continual learning results on Split CIFAR100. Top two rows are multi-head, bottom two rows are single-head.

# H   Additional Active Learning Results

In Section 5 we provided active learning results for CNAPs and Prototypical Networks on the VGG Flowers dataset and three held out test languages from the Omniglot dataset. Here, we provide the results from all twenty held-out languages in Omniglot.

Figure H.13 demonstrates that in almost all held-out languages, using the predictive distribution of CNAPs not only improves overall performance, but also enables the model to make use of standard acquisition functions [35] to improve data efficiency over random acquisition. In contrast, we see that in most cases, random acquisition performs as well or better than acquisition functions that rely on the predictive distribution of Prototypical Networks. This provides empirical evidence that in addition to achieving overall better performance, the predictive distribution of CNAPs is more calibrated, and thus better suited to tasks such as active learning that require uncertainty in predictions.

Figure H.13: Active learning results on all twenty held-out OMNIGLOT languages.

## Footnotes

[3]https://pytorch.org/docs/stable/torchvision/models.html