[Reviews · NeurIPS 2019]

Reviewer 1



In this paper, the authors proposed a multi-task learning approach for few-shot learning using deep neural networks. The proposed method can automatically adapt to new tasks at testing time after initial multi-task training. To make this possible and achieve better performance over existing methods, the authors proposed to use two networks to learn task-specific parameters in the classifier. One is for task-specific parameters in the final classification layer. The other one is to learn task-specific parameters that adapt the common feature extractor (a network) shared among tasks to be task specific. The superior of this method is demonstrated by not only the better than the-state-of-the-art performance on few-shot learning problems, but also competitive performance on continual learning and active learning tasks. Even though there have been several existing works on few-shot learning, as demonstrated by the empirical results in the paper, this work significantly moves the-state-of-the-art. The paper is well organized and easy to follow. I found the illustrations, i.e., Figure 1-3 are very helpful for me to understand the architecture of the proposed method. There is just one typo that I noticed on line 246, D_{\tau}, should it be D_^{\tau} to keep it consistent with the notations in the rest of the paper?

Reviewer 2



Quality: The technical content of the paper is well motivated and the approach taken is interesting. However, a few things are worth mentioning. 1 - The classification parameters for a given class are generated independently from the other classes. This means that the classifier is more likely to act as a prototypical model than a discriminative one. 2 - In the adaptation network, the auto-regressive component is not technically motivated. The fact that it improves results just shows the lack of capacity in the FiLM network as a way to modulate the feature extractor parameters alone. Did you compare different ways of modulating the feature extractor parameters? 3 - z_G is computed using only the inputs from the query set, what about the labels? 4 - The statement “ Allowing θ to adapt during the second phase violates the principle of “train as you test", i.e., when test tasks are encountered, θ will be fixed, so it is important to simulate this scenario during training “ is technically false as within each meta-learning step θ will be fixed even when is not pretrained. Thus, the justification for the training procedure is a bit weak despite the comparison between the proposed approach and the classical one. Maybe the sensitivity of the hyper-parameters is more the main reason for those differences. 5 - Related to the previous point, pretraining θ requires a large dataset, which is not always available in other domains as it is in computer vision, do not play in favor of the proposed training procedure. Thus, it is critical to find an alternative that works for training all parameters together using the meta-dataset instead of the two-phase approach proposed. 6 - Despite great results shown for the few-shot learning settings, the results section is a bit unfocused as the application to active learning and continual learning seems unnatural and forced. Clarity: The paper is generally well-written and structured and really easy to understand. Originality: The main originality in this work is definitely the auto-regressive modulation network that was proposed. Significance: This work shows significant improvements over the state of the art in few-shot classification which is an important contribution. While weakly motivated, it also proposes a new neural net architecture that improves upon modulation results achieved by FiLM, which helps to achieve better results.

Reviewer 3



The authors present a method called Conditional Neural adaptive Processes (CNAPs) able to efficiently solve new multi-class classification problems after an initial pre-training phase. The proposed approach, based on Conditional Neural Processes[1], adapts a small number of task-specific parameters for each new task encountered at test time. These parameters are conditioned on a set of training examples for the new task, don't require any additional tuning and adapt both the final classification layer and the feature extraction process, allowing to handle different input distribution. While being very close to CNP, this work focuses on the image classification task and makes several addition to the original method. These additions (FiLM layers, auto-regressive feature adapter, usage of deep sets) are clearly justified and their individual contributions are explored in the different experiments. The major negative point of this paper is its similarity with CNPs. The authors compare the two approaches in section 2 (lines 67-70), but this argument is not convincing at all, the adapted parameters can also be seen as a simple vector. I think the article would gain from putting a bigger emphasize on the auto-regressive way of dynamically adapting the parameters, which is an interesting and novel contribution. The article is very well written. While the approach is complex, the authors did a good job at progressively presenting the different components used, with clear explanations and corresponding references to justify each choice they made. [1] Conditional neural processes. Garnelo et. al. 2018

[Author Response · NeurIPS 2019]

We thank all the reviewers for their constructive and useful feedback! Below, we address the key comments in order.

**R2: Classification parameters are independent of other classes, similar to prototypical models.** We thank the reviewer for pointing out this important issue. The classifier component of the inference network is similar in approach to prototypical models, and in fact, [5] shows that prototypical networks are recovered as a special case. Surprisingly, this approach is optimal in certain circumstances, and there is a principled justification for this design choice. This is best understood through the lens of density ratio estimation [i, ii], which shows that an optimal softmax classifier learns the ratio of the densities: $\text{Softmax}(y = k|x) = p(x|y = k)/\sum_j p(x|y = j)$, assuming equal a priori probability for each class. Our system follows this optimal form by setting: $\log p(x^*|y = k) \propto h_\theta(x^*)^T w_k$, where $w_k = \psi_\phi(\{x_n\})$ for each class in a given task. Here $\{x_n\}$ are the few-shot context examples for class $k$, and $x^*$ is the target example. This argument states that under ideal conditions (i.e., we can perfectly estimate $p(y = k|x)$), the context-independent assumption is correct, motivating our design. A discussion on this point is in [5], Appendix B.1.

**R2: AR component is not technically motivated. Comparison to additional forms of modulation?** To adapt layer $l$, the system must have access to the representation of task relevant inputs at layer $l - 1$. While $z_G$ will encode how layer $l - 1$ has adapted, it is useful to directly observe the representation of the context set at layer $l - 1$. This is similar to the linear classifier adaptation, which leverages the task-specific feature representation. We indeed performed a study comparing FiLM layers with parallel residual adapter methods [17] (which have significantly more capacity) using the non-AR variant, with no gain in performance. These experiments indicate that the AR approach is superior to naively adding capacity to the adapters. We will provide results from this ablation study as suggested.

**R2: $z_G$ is computed using only the inputs from the query set, what about the labels?** We agree that performance could potentially be improved by utilizing the labels from the context set (e.g., MAML [7] and LEO [23]). In our design, we took inspiration from unsupervised pre-training approaches to adapting feature extractors. This is arguably more limited but has advantages e.g. it can be used for semi-supervised learning (with the unlabeled examples used to adapt the feature extractor). However, this is indeed an avenue for future exploration.

**R2: The justification for the training procedure is weak...** We find that it is crucially important for the inference networks to *learn to adapt* the feature representation at test time. Joint training results in a network that captures the training data making minimal use of the adapters, as the feature extractor is very flexible and learns features that are suitable to solve each training task with limited adaptation. This network generalizes poorly since it has not experienced data sets for which the feature extractor requires significant adaptation. The two stage training fixes this issue by forcing the adaptation to occur in the second stage just as it must at test time. An ablation study is provided in Table D.4, demonstrating significant differences between the two approaches. We will expand this discussion point in the revision.

**R2: Pretraining $\theta$ requires a large dataset, which is not always available.** We agree that this is an assumption / limitation of our method. However, in this work we are mainly concerned with the vision domain, where pre-trained networks are readily available, and indeed, it is necessary to leverage them to achieve SoTA performance. The assumption of large related corpora carries to other interesting domains e.g., speech, NLP, and recommender systems.

**R2: Results section is a bit unfocused. Experiments on active and continual learning seem forced.** Continual learning was a major point of motivation in this work. We will improve the writing so that that the exposition better reflects this, as it shaped many of the design decisions. Further, we wanted to demonstrate the flexibility of our approach; continual learning results show that our method outperforms a very strong base line with far fewer shots (which we see as an important result), and the active learning results suggest well calibrated uncertainties.

**R3: The major negative point of this paper is its similarity with CNPs.** We fully agree that CNAPs *is* an extension of CNPs, with design choices targeted towards multi-task classification and continual learning. In particular, CNPs cannot handle varying-way tasks (see [13] Sec. 4.3) as required for the meta-dataset task and continual learning. Further, increasing (pre-specified) way results in (at least) linear growth of parameters in the decoder. In contrast, CNAPs handles varying-way tasks on the fly with a fixed number of parameters via the classifier adaptation network. We emphasize further differences: (i) CNAPs employs a parameter sharing hierarchy (global / task / class); (ii) $\psi^\tau$ directly parameterizes the feature mapping as opposed to being a fixed-dimensional input to the decoder (as in CNPs); (iii) the dimension of $\psi^\tau$ increases with the number of classes in $\tau$; and (iv) CNAPs employs a meta-training procedure geared towards *learning to adapt* to diverse tasks. We will improve the discussion and add an ablation study for point (ii).

**R3: I think the article would gain from putting a bigger emphasize on the auto-regressive adaptation.** We agree that the AR adaptation is under-emphasized, and will add detail and emphasis on this aspect of the model.

**Additional References (numbered references correspond to main paper bibliography)**

[i] S. Mohamed. The Density Ratio Trick. The Spectator (Blog). 2018

[ii] M. Sugiyama, T. Suzuki, and T. Kanamori. Density ratio estimation in machine learning. 2012.


[Meta-Review · NeurIPS 2019]

This paper proposes a novel few-shot learning method, with a specific application focus to fine-tuning CV object classification models from pre-trained features. Different from previous few-shot learning or CNP work, this work tries to address a convincing real world use case. Its novelties include inference amortization for head models, adaptation of the feature network on each task using a novel autoregressive architecture. One point of improving the paper is to move details about the autoregressive model structure and the adaptation network into the main text. Too many relevant details are just in the supplemental material.